# Molecular Diagnostics in Human Papillomavirus-Related Head and Neck Squamous Cell Carcinoma

**DOI:** 10.3390/cells9020500

**Published:** 2020-02-22

**Authors:** Katherine C. Wai, Madeleine P. Strohl, Annemieke van Zante, Patrick K. Ha

**Affiliations:** 1Department of Otolaryngology Head and Neck Surgery, University of California San Francisco, San Francisco, CA 94158, USA; katherine.wai@ucsf.edu (K.C.W.); madeleine.strohl@ucsf.edu (M.P.S.); 2Department of Pathology, University of California San Francisco, San Francisco, CA 94158, USA; annemieke.vanzante@ucsf.edu

**Keywords:** human papillomavirus (HPV), head and neck cancer, p16

## Abstract

The incidence of human papillomavirus (HPV)-related head and neck squamous cell carcinoma continues to increase. Accurate diagnosis of the HPV status of a tumor is vital, as HPV+ versus HPV– tumors represent two unique biological and clinical entities with different treatment strategies. High-risk HPV subtypes encode oncoproteins E6 and E7 that disrupt cellular senescence and ultimately drive tumorigenesis. Current methods for detection of HPV take advantage of this established oncogenic pathway and detect HPV at various biological stages. This review article provides an overview of the existing technologies employed for the detection of HPV and their current or potential future role in management and prognostication.

## 1. Introduction

The incidence of human papillomavirus (HPV)-related head and neck squamous cell carcinomas (HNSCCs) has been increasing in recent decades [1,2]. HPV-related tumors arise most frequently in the oropharynx, the narrow ring of lymphoid-rich tissues that include the tonsils, base of tongue and soft palate. HPV+ cancer of the oropharynx represents an epidemiologically, biologically, and clinically distinct entity compared to its HPV– counterpart and is associated with a better overall prognosis [3,4]. Given this difference in prognosis and implications for treatment, it is paramount to precisely ascertain the HPV status of a tumor with precision.

High-risk subtypes of HPV are known to be tumorigenic in cervical cancer, and accepted to be largely similar in oropharyngeal cancer, though the exact mechanism remains an area of active research [5]. HPV16 is the predominant subtype implicated in tumors of the head and neck and, to a lesser extent, HPV18. High-risk HPV subtypes can integrate into the human genome (Figure 1). In doing so, viral oncoproteins E6 and E7 are overexpressed and these proteins promote tumor progression by inactivating the *TP53* and retinoblastoma tumor suppressor gene products, respectively [6,7]. The E7 viral oncoprotein binds to retinoblastoma protein (Rb), disrupting the cell cycle and, ultimately, initiating the transcription of S-phase genes. This G1 to S phase of the cell cycle is in part controlled by the interaction of p16 with Rb [8]. In HPV-driven carcinogenesis, Rb is functionally absent and p16 is overexpressed due to the loss of negative feedback [9]. In contrast, the majority of non-HPV-related HNSCCs have a disruption of *TP53*, resulting in cell cycle dysregulation in the absence of p16 upregulation [10,11].

The simplest methodology for the detection of HPV takes advantage of this distinct oncogenic pathway and uses the upregulation of p16 expression as a surrogate for high-risk HPV [12]. HPV-specific tests include viral DNA detection by polymerase chain reaction (PCR) or in situ hybridization (ISH) or HPV RNA detection by reverse-transcription PCR or ISH. Some of these methodologies can be applied not only to tissue specimens, but also fine-needle aspiration biopsy (FNAB) specimens, saliva and serum samples. In this review, we provide an overview of the existing technologies for the detection of HPV+ HNSCCs and their current or potential roles in clinical diagnostic and prognostic applications.

## 2. Detection of HPV in Tissue Biopsies

### 2.1. p16 Staining of Tissue Specimens

Immunohistochemical (IHC) staining of p16 is an excellent and accepted surrogate marker for HPV in oropharyngeal squamous cell carcinomas (OPSCCs). Recent guidelines from both the American Society of Clinical Oncology (ASCO) and the College of American Pathologists (CAP) recommend that all oropharyngeal tissue specimens undergo testing for high-risk HPV status, and p16 testing should be performed first, prior to HPV-specific testing [13,14]. Similarly, both guidelines state that HPV testing by p16 IHC should be routinely employed in SCCs of unknown primary metastatic to the upper and middle cervical lymph node chains (levels II and III). Although not previously standardized, the guidelines recommend that p16 positivity should be defined as >70% of tumor cells showing moderate–strong nuclear and cytoplasmic staining. A recent systematic review of pooled data found that among OPSCC patients, p16 IHC has a sensitivity of 94% (95% CI 91–97%) and specificity of 83% (95% CI 78–88%) [15].

The clinical implication of discordant cases (i.e., cases which are HPV– by p16 IHC, but HPV+ by PCR or ISH) remains an ongoing concern [16]. Discordant cases, which can be up to 17% of OPSCCs, may reflect cancers that harbor HPV that are not transcriptionally active, a “bystander” virus from adjacent benign mucosa or entrapped saliva, or a different biological entity altogether. Cases with discordant p16 and HPV-specific tests have been shown to have a distinct prognosis. For example, two similar meta-analyses demonstrated improved outcomes with 5 year disease-free and overall survival in those with p16+/HPV+ OPSCCs compared to those with either p16-/HPV+ or p16+/HPV– [17,18]. Interestingly, the prevalence of discordant cases increases when HNSCCs arising from subsites other than the oropharynx are tested. Thus, upregulation of p16 is not an acceptable surrogate for HPV in subsites other than the oropharynx, as the prognostic implications of p16 positivity in non-OPSCCs has not been established [19,20].

Despite some limitations, p16 status alone among patients with OPSCCs is an important, independent prognostic biomarker [13,21,22]. Additional studies regarding the prognostic utility of p16 IHC and HPV-specific testing in other head and neck subsites are required to understand its significance.

### 2.2. HPV ISH in Tissue Specimens

It has been suggested that in order to be biologically and clinically relevant, HPV must be transcriptionally active. Thus, some suggest that detection of HPV RNA should be the gold standard for HPV testing. For example, patients who are HPV DNA+/RNA+ have improved overall survival compared to those who are HPV DNA+/RNA– [23]. RNA ISH can be used to detect viral E6/E7 mRNA [24] and allows visualization of transcriptionally active virus within tumor cells. RNA ISH has increased sensitivity compared to DNA ISH, likely reflecting the fact that although viral DNA can be present in low copy numbers, transcription is a natural amplification step that results in high levels of viral mRNA. Depending on the DNA ISH method, DNA ISH fails to detect HPV in 16–20% of cases in which RNA ISH is positive [25]. A separate study compared p16 IHC, DNA ISH, manually performed RNA ISH, and automated RNA ISH on head and neck surgical specimens. They used detection of HPV by PCR as the gold standard. The manual RNA ISH platform was 91% sensitive compared to 65% for DNA ISH. RNA ISH had strong correlation with p16 status (kappa 0.9) and PCR (kappa 0.8). Furthermore, in this study, an automated RNA ISH protocol was compared to a manual protocol and found 96% concordance. Overall, RNA ISH had improved sensitivity and similar specificity when compared to DNA ISH [26]. A separate validation study of a different, commercially available RNA ISH detection system showed a sensitivity of 97% and specificity of 93% when compared to quantitative reverse transcription PCR as the gold standard. Moreover, the authors of this later study demonstrated that HPV status by RNA ISH had independent prognostic significance [27]. As RNA ISH appears to be the most promising HPV-specific test currently available, efforts should be made to implement standardized automated RNA ISH testing for clinical use in all OPSCCs.

### 2.3. HPV PCR in Tissue Specimens

Multiple PCR assays exist for detection of HPV DNA in tissue specimens. PCR assays were originally developed for cervical samples, and can detect multiple types of HPV, including high-risk HPV subtypes [28]. This methodology can detect short, fragmented DNA, which often occurs after formalin fixation [29]. Detection of HPV DNA by PCR has been shown to have prognostic significance similar to p16 status. Those found to have HPV+ oral cavity/oropharynx cancer by PCR have improved overall and disease-specific survival compared to those with HPV– cancer [30,31]. However, there are two major concerns with using PCR for detection of HPV DNA. The first is that the test may be overly sensitive; it is impossible to distinguish whether the HPV DNA detected by this method is from the malignant tumor tissue or “bystander” virus present in adjacent, non-neoplastic tissues. Multiple studies have shown detectable HPV DNA in tumor-free tissue [32,33]. The second concern is that detection of HPV DNA by PCR, similar to DNA ISH, does not provide information on whether the virus is transcriptionally active. To address this later issue, a few studies have shown that reverse transcription PCR may be used on formalin-fixed specimens to detect transcripts of E6 and E7 mRNA, although this method is not currently available for clinical application, and has not been compared to other methodologies [34,35].

## 3. Detection of HPV in Fine-Needle Aspiration Biopsy Specimens

No definitive criteria exist for determination of HPV status from fine-needle aspiration biopsies (FNABs). The ASCO and CAP guidelines recommend that all FNA samples of SCCs of oropharyngeal origin and SCCs of unknown primary should be tested for HPV status; however, no strong recommendation was made for the method of detection [13,14].

### 3.1. p16 Staining of FNA Biopsies

A number of studies have been performed to determine the utility of p16 staining of FNABs. Unlike tissue specimens where the cut-off for p16 positivity has been defined as >70% of tumor cells showing moderate–strong nuclear and cytoplasmic expression, there is no standard for FNAB specimens. Several studies suggest that the cut-off of 70% for FNAB samples is too high, generating many false-negative results. To address this question, a recent study was undertaken to determine optimal test characteristics for p16 immunocytochemistry (ICC), utilizing p16 status on corresponding surgical samples as the gold standard [36]. Using receiver-operating characteristic curves in their results, the authors propose using a 50% threshold for p16 ICC on FNAB samples, which was associated with a sensitivity of 74% and specificity of 100%, compared to the 70% threshold with a sensitivity of 45% and specificity of 100%. However, other studies that examine p16 IHC on FNAB support much lower cut-offs of 10–15% [37,38]. Jalaly et al. demonstrated high concordance rates between p16 IHC and RNA ISH, with a cut-off of 15% [37]. Even at a threshold of 10%, concordance between p16 IHC on FNAB, p16 IHC on tissue specimens, and HPV DNA ISH performed on FNAB specimens was high (kappa 0.65 and 0.71 respectively) [38].

Buonocore et. al. recently suggested that the variation in the optimal cut-off for FNAB specimens reported in the literature could be due to differences in fixation and preparation methods utilized, as FNAB specimen processing is not standardized across institutions [39]. In their study, they compared p16 IHC staining patterns between cell blocks prepared from specimens preserved in CytoLyt (an ethanol-based fixative) and cell blocks prepared from specimens fixed in formalin. They found that formalin-fixed cell blocks displayed diffuse, consistent p16 staining compared to CytoLyt-fixed material, which had weaker staining, hampering accurate interpretation.

Overall, it is clear that the cut-off for p16 positivity in FNAB specimens should be lower than that established for tissue specimens. There is currently no recommendation for what the cut-off should be and, given the absence of a standardized methodologies for cell block preparation, individual laboratories must validate p16 staining on cytologic specimens against paired tissue specimens. Despite high concordance rates and overall good specificity for p16 IHC on FNAB specimens, the CAP guidelines suggest that p16 IHC and/or HPV-specific testing should be repeated if p16 is negative on an FNAB specimen and additional tissue becomes available.

### 3.2. HPV ISH on FNA Biopsies

DNA ISH can be utilized in detection of HPV in cytologic samples—though similar to other HPV testing modalities applied to FNAB, its role in clinical practice is not well defined. Most studies that address DNA ISH correlate DNA ISH with p16 staining on cell blocks, rather than commenting on its role in clinical practice. Although the gold standard for HPV detection and the definition of p16 positivity in FNABs vary across studies, DNA ISH and p16 IHC performed on FNAB samples have high concordance rates [38,40,41]. Importantly, one study demonstrated few discrepancies between p16/HPV DNA ISH status on FNAB samples compared to p16/HPV testing of the subsequent tumor specimen [40].

Similar to HPV DNA ISH in FNAB samples, there is limited data regarding the clinical role for HPV RNA ISH in FNAB specimens. A positive test result has not been defined, and it is not currently accepted as a confirmatory test. Using PCR on DNA extracted from cell block as the gold standard, RNA ISH on cell blocks demonstrates high sensitivity at 97% and has a 93% concordance rate with cytologic p16 positivity status (regardless of the cut-off for a positive test) [42]. A different study observed the same concordance with a cut-off of >15% for p16 positivity in cell block [37].

### 3.3. HPV PCR on FNA Biopsies

Detection of HPV by PCR in FNAB samples is feasible, though its role in clinical practice has yet to be established. Many different assays with different sets of primers exist to facilitate detection. Using various assays, studies report high concordance rates between HPV detection by PCR on FNABs with p16 and DNA ISH on tissue specimens, although these studies did not include FNABs from HPV– lesions to establish specificity [43,44,45,46]. Channir et. al. validated PCR on FNAB specimens. In this study, PCR was run on FNABs from metastatic OPSCCs and oral cavity SCCs; FNABs from branchial cleft cysts and Warthin’s tumors served as negative controls [47]. Using p16 IHC on tissue specimen as the gold standard, the authors concluded that PCR had a sensitivity of 94.7% (95% CI, 82–99%) and specificity of 100.0% (95% CI, 93–100%). For comparison, detection of HPV by PCR in the context of cervical cancer screening performs similarly, with a sensitivity of 92–99% and specificity of 82–99%, depending on which PCR assay is employed [48].

## 4. Detection of HPV in the Blood

### 4.1. Detection of HPV DNA in Blood Samples

The interest and research in circulating tumor cells (CTCs) and circulating cell-free tumor DNA (ctDNA) has increased dramatically over the past decade, as the idea that a serum sample or “liquid biopsy” could allow for screening and surveillance in a non-invasive fashion. CTCs can be released into the bloodstream by a primary tumor or metastases. The majority of ctDNA is from apoptotic or necrotic tumor cells, which subsequently release fragmented DNA into circulation.

As expected, patients with HPV+ tumors were more likely to have detectable serum levels of HPV DNA by real-time PCR compared to HPV– tumors pre-treatment [49]. Furthermore, the detection of HPV DNA in serum is likely independent of the overall concentration of cell-free DNA [50]. A systematic review by Jensen et. al. included five different studies concerning plasma detection of HPV DNA, and reported a pooled sensitivity of 54% (95% CI 30–75%) and pooled specificity of 98% (95% CI 93–99%) [51]. More recent studies have used droplet digital PCR, a technology with similar sensitivity to standard PCR but with improved accuracy, and applied this to ctDNA [52,53]. Recently, one study compared plasma samples from four different patient populations: (1) HPV+ OPSCCs or HPV+ anal cancer, (2) HPV– HNSCCs, (3) HPV+ OPSCCs who had completed definitive treatment, and (4) patients without cancer [54]. All patients had samples collected pre-treatment and, for a subset, additional samples were collected weekly during and after definitive chemoradiation. At baseline, no ctDNA was detected among the two negative control groups, specifically those with HPV– HNSCCs and those without cancer, implying 100% specificity. Of 97 patients with HPV+ OPSCCs, 90 had detectable HPV16 ctDNA pre-treatment, which corresponds to 92% sensitivity. Among the subset of patients who were followed during treatment, HPV16 ctDNA cleared by week 7 in the majority of patients, suggesting this technology holds promise in assessment of treatment response and surveillance. Another study from Chera et. al. demonstrated similar results, with a sensitivity of 89% and specificity of 97% for HPV16 ctDNA [55].

### 4.2. Circulating HPV DNA as a Biomarker

While ctDNA shows promising results in terms of detection, its role as a biomarker is not well defined, and the current literature demonstrates conflicting results. In a subset of 99 patients with HPV+ OPSCC tumors, one study found no difference in disease-free survival at 3 years between patients with regard to pre-treatment serum HPV DNA status (as determined by real time PCR amplification of HPV16 E6 and E7 region) [49]. Furthermore, the data for post-treatment follow-up blood samples were mixed as not all patients with HPV+ tumors had detectable levels of serum HPV DNA prior to locoregional recurrence. In contrast, when using quantitative reverse transcription PCR to detect HPV16 E6/E7 expression, the presence of circulating HPV16 E6/E7 mRNA at baseline, prior to treatment, was associated with worse progression-free survival and overall survival [56]. This was similarly true of different individual studies, which found that increased levels of pre-treatment HPV16 ctDNA were correlated with worse prognosis [57], and in one study, worse TNM staging as well [58].

Separately, another group designed and validated a novel HPV DNA next generation sequencing (NGS) technique for plasma, and was able to demonstrate its utility during surveillance among patients with locally advanced HPV+ HNSCCs [59]. When compared to HPV RNA quantitative PCR performed on tissue biopsy as the gold standard, the baseline serum HPV DNA NGS technique had a sensitivity of 93% and specificity of 100%. After completing chemoradiation therapy (CRT), 23 of 27 patients had complete radiologic response by PET-CT at 12 weeks post-treatment and undetectable serum levels by NGS. Of those without complete radiologic response, 3 of 4 had an FDG-avid lesion at the primary tumor site and undetectable serum levels, and none had pathologic evidence of residual tumor on biopsy. One patient had ongoing detectable serum HPV DNA and was ultimately diagnosed with recurrence in cervical lymph nodes eight months after definitive CRT. Another study using conventional and real-time quantitative PCR rather than NGS showed similar results, with the majority of patients demonstrating undetectable levels of serum HPV DNA and complete clinical response [60]. Of the three patients that did experience recurrence in this later study, all had detectable levels of HPV DNA at the time of recurrence.

In a different study, Chera and colleagues analyzed the impact of both the rate of clearance and the number of copies of HPV16 ctDNA (using digital droplet PCR) on prognosis [55]. Based on previously published data for Epstein–Barr virus-related nasopharyngeal cancer, rapid clearance was defined as >95% clearance of HPV16 ctDNA by week 4 of CRT. All patients with rapid clearance had a complete clinical response. Furthermore, they demonstrate that patients with ≤200 copies/mL of HPV16 ctDNA pre-treatment have adverse tumor genetics, specifically lower tumor HPV copy number and increased integration of HPV DNA into the somatic genome. Taken together, patients with favorable HPV16 ctDNA profile (>200 copies/mL and rapid clearance) had 100% recurrence/disease-free survival at 2 years, compared to those with an unfavorable profile (≤200 copies/mL or failing rapid clearance).

Overall, the current literature supports that serum HPV DNA may be a useful prognostic biomarker during and after treatment and may be helpful in monitoring treatment response. However, extrapolating from these studies to create a practical and standardized approach to clinical surveillance is challenging, as the technologies utilized are not widely available in the clinical setting. Clearly, more studies with longer-term follow-up are required.

### 4.3. Circulating Antibodies against HPV

Multiple studies have shown that the presence of IgG antibodies to HPV16 is associated with increased odds of having HPV+ OPSCCs, which has helped to confirm its etiologic role [61,62]. Given that these antibodies can be detected in the blood, several studies have been performed to determine the utility of non-invasive screening or monitoring with antibody levels. Some have expanded their methodology to encompass IgM, IgA and IgG with more expansive assays [63]. A large population-based study in Europe compared pre-diagnosis plasma levels of several HPV antibodies in cases of HNSCCs versus esophageal cancer and healthy controls [64]. Patients with OPSCCs specifically were more likely to be HPV16 E6 seropositive compared to healthy controls (OR = 274; 95% CI, 110 to 681), and this relationship remained true up to 10 years before the diagnosis of cancer was made. However, this study did not collect HPV status of the tumor, and thus we are unable to draw conclusions specific to HPV+ cancers. In terms of diagnostic testing for patients with OPSCCs, E6 seropositivity demonstrated the best test characteristics with a sensitivity of 96% (95% CI 88–98%) and a specificity of 98% (95% CI 90–100%). This is comparable to other commonly accepted techniques, including p16 IHC [65].

The role of serology in surveillance and/or prognosis has also been examined. An early study from Rubenstein et. al. combined all patients with primary HNSCCs and showed that the HPV antibody titers decreased when comparing baseline to post-treatment levels, though this was not stratified by tumor site or HPV status of the tumor itself [66]. More recently, another study was conducted among patients who had HPV+ tumors by ISH, in which the authors quantified baseline and follow-up levels of antibodies to HPV E6 and E7 [67]. Both antibody levels decreased when comparing pre-treatment levels to post-treatment levels. This study additionally showed that increased baseline E6 antibody levels have prognostic significance and, even after adjustment, remain independently associated with recurrence (HR 7.1, 95% CI 1.2–43.2). A separate study also showed that antibody levels among HPV+ SCCs of unknown primary decrease with treatment [68]. However, no study has commented on the significance of following antibody levels over time, and whether serology may be used to identify recurrence before it becomes clinically manifest. Thus, while there are data to suggest antibody levels may be both a diagnostic and prognostic biomarker, additional longitudinal studies are required.

## 5. Detection of HPV in Saliva

Multiple studies have been undertaken to assess the feasibility and utility of detection of high-risk HPV in salivary samples in hopes of developing a screening tool for HPV+ HNSCCs or a non-invasive surveillance test following treatment.

### 5.1. Salivary HPV DNA

In a cohort of 150 healthy, adult participants, 2.6% (4/151) were found to harbor HPV16 DNA using PCR on saliva, while none had HPV18 DNA [69]. A study by Zhao et. al. compared the detection of HPV by quantitative PCR in saliva among both healthy controls and patients with known head and neck SCCs [70]. The DNA probes utilized were specifically designed to amplify the E6 and E7 regions of HPV16. HPV in the saliva was detectable in 50% of patients with HPV16 positive tumor samples, 18% of patients with HPV16 negative tumor samples, and 2.7% of healthy controls. Depending on the cut-off value for HPV copy number, the sensitivity of their oral rinse HPV test ranged from 14% to 32% and the specificity from 97% to 99%. Overall, they suggest that while the test in its current form would be difficult to apply to broader populations, identification of high-risk groups may allow for clinical application. A separate, more recent study analyzed the diagnostic accuracy of detection of HPV in saliva in patients with known HPV+ HNSCCs [71]. Tumor HPV status was determined by p16 IHC and/or HPV DNA ISH on tumor samples and PCR amplification of 37 different subtypes of HPV was performed on oral rinses. The authors concluded that the oral rinse test had a sensitivity of 72.2% and a specificity of 90%, similar to prior studies [72]. One publication has shown conflicting results regarding the utility of HPV detection in saliva, but definitive conclusions may be difficult to draw due to small sample size in that case-control study [73].

Interestingly, there is observational data that salivary HPV detection may be used in surveillance. One study examined salivary samples from patients with HPV+ OPSCCs and found that the trend in salivary HPV DNA levels corresponded to treatment response [74]. Saliva samples from a cohort of 59 patients with HNSCCs were analyzed at multiple time points [75]. Saliva was taken prior to initiation of any treatment, at 3 months after definitive treatment, and every 3 months for a total 2 years. In total, 20 of 59 patients had HPV+ SCCs, and four developed recurrence. Using quantitative PCR for detection of HPV E6 and E7, two of these four had detectable HPV in their saliva with a lead time of 3.5 months. Although limited by sample size, these findings suggest there may be a role for HPV detection in saliva for surveillance. Similarly, in another small study of patients with HPV+ SCCs, the authors found that among nine patients undergoing surveillance by saliva testing, three patients had HPV detected in their saliva before clinical evidence of recurrence and one had HPV detected with a high probability of recurrence (but not yet biopsy proven at the time of publication). The remaining five patients did not have HPV in their saliva and no recurrence [76].

A few studies exist that address the utility of both saliva and plasma detection of HPV DNA. One found that the sensitivity and specificity of post-treatment saliva HPV16 status in predicting recurrence within 3 years were 19% and 97% respectively in patients with HPV16 positive tumors. The sensitivity and specificity of post-treatment plasma HPV16 status was 55% and 96% respectively [77]. However, when post-treatment saliva and plasma HPV16 status were combined, this increased to 70% sensitive and 91% specific. Similar results were seen in a different study in which HPV was detected in 40% of saliva as opposed to 86% of plasma samples [76].

### 5.2. Salivary microRNA

Extracellular microRNAs (miRNAs) have also been studied as possible diagnostic and prognostic biomarkers, but few studies have applied this to saliva samples. The majority of current studies are limited to oral cavity carcinoma, without differentiation between HPV+ and HPV– tumors [78,79,80]. These studies do suggest that miRNA detection can differentiate between those with and without HNSCCs, and that miRNA levels in saliva are significantly altered in the post-operative or post-radiation period, which suggests that salivary miRNA may have a role in surveillance. More recently, Wan and colleagues studied both individualized miRNA expression levels as well as multi-marker panels. They showed that different miRNA panels can distinguish HPV– patients from healthy controls (sensitivity 60%, specificity 94%) and HPV+ patients from healthy controls (sensitivity 65%, specificity 95%) [81]. Furthermore, they found that elevated levels of miRNA-210 could distinguish HPV+ from HPV– patients.

Together, the literature suggests that while the identification of HPV in saliva is feasible, its role in screening, diagnosis and prognosis is not yet well defined. In terms of screening, the entire adult population would have a false positive rate too high for the expense of salivary testing; however, if a higher risk population was identified, those individuals may benefit from screening. More immediately, HPV detection in saliva may have a role in surveillance, although larger, prospective cohort studies are required prior to implementation.

## 6. Conclusions

Multiple methodologies exist for the detection of HPV in HNSCCs. These are most relevant to tumors arising in the oropharynx, where HPV status has significant prognostic implications. A summary of HPV detection methods is presented in Table 1, though it is challenging to compare test characteristics, as there remains no currently accepted gold standard. Some of these methodologies are now routinely employed clinically, such as p16 IHC for OPSCCs. Though promising, many tests remain in the research realm. Newer technologies for detection of HPV in serum and saliva provide exciting avenues for non-invasive treatment monitoring and surveillance, but their broader application is currently limited by feasibility and limited longitudinal follow-up Table 1.

## Figures and Tables

**Figure 1 cells-09-00500-f001:**
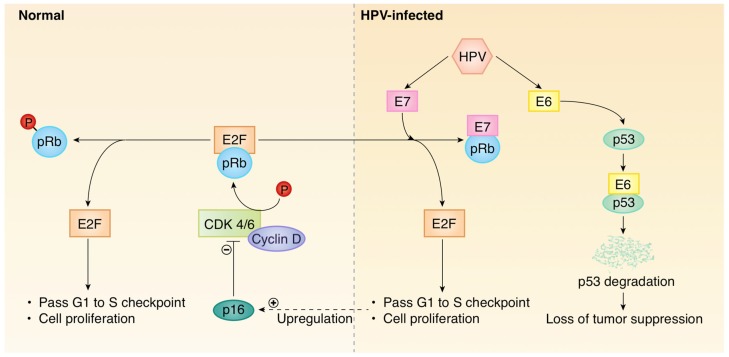
Human papillomavirus (HPV)+ cancer increases expression of p16. Left panel: Normal, uninfected cell. Cyclin D–cyclin dependent kinase (CDK) 4/6 complex initiates phosphorylation of the tumor suppressor protein, pRb. The hyperphosphorylation of pRb leads to release of the transcription factor E2F into its active state, which drives the expression of downstream gene products allowing the cell to transition from the G1 to S phase. As a cyclin kinase inhibitor, p16 is a tumor suppressor and negative regulator of the cyclin D–CDK 4/6 complex. Right panel: HPV infected cell. When the transcription factor E2F is bound to pRb, it remains inactive. The overexpression of the E7 oncoprotein by high-risk HPV subtypes disrupts the E2F–pRb complex by displacing E2F and binding to pRb. The subsequent release of E2F into its active state drives the expression of downstream gene products, allowing the cell to transition from the G1 to S phase. In a regulatory feedback attempt to inhibit further cell proliferation, p16 is upregulated, and thus can be a surrogate for HPV+ tumors. The overexpression E6 oncoprotein acts via a separate mechanism. E6 binds to the tumor suppressor protein, p53, and ultimately leads to degradation of p53. Loss of the regulatory function of p53 causes aberrant propagation of the cell cycle and prevents apoptosis.

**Table 1 cells-09-00500-t001:** Test characteristics for detection of HPV in head and neck squamous cell carcinomas (HNSCCs).

Specimen Source	Diagnostic Test	Sensitivity	Specificity	Reference
Tissue specimen				
	p16 IHC	94%	83%	Prigge et al. [15]
	DNA ISH	65%	94%	Kerr et al. [26] Schache et al. [27]
	RNA ISH	91–97%	93–94%	Kerr et al. [26]
	DNA PCR	97%	87%	Schache et al. [82]
Fine-needle aspiration				
	p16 IHC	39–70% *	100%	Yang et al. [36] Xu et al. [38]
	DNA ISH	-	-	-
	RNA ISH	97%	-	Wong et al. [42]
	PCR	94%	100%	Channir et al. [47]
Blood/serum				
	Circulating tumor DNA by PCR	54–92%	97–100%	Jensen et al. [51] Damerla et al. [54] Chera et al. [55]
	E6 antibody (oropharynx)	96%	98%	Holzinger et al. [65]
	E6 antibody (other subsites)	50%	100%	Holzinger et al. [65]
Saliva				
	DNA PCR	72–79%	90%	Qureishi et al. [71] Rosenthal et al. [72]
	microRNA	65%	95%	Wan et al. [81]

* Sensitivity varies based on cut-off used for positive p16 result. IHC, immunohistochemical staining; ISH, in situ hybridization.

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
