# Peer review of "Molecular Diagnostics in Human Papillomavirus-Related Head and Neck Squamous Cell Carcinoma"

_cells, 2020, doi:10.3390/cells9020500_

Round 1
Reviewer 1 Report
This is a useful and well-written review on diagnostic approaches to HPV-positive oropharyngeal cancers. The main approaches to diagnosis have been well described in a critical manner. There is an extensive and very useful reference list.
However, there are a few minor changes that could make the molecular aspects of the review more accurate. Indeed the molecular basis of HPV-associated OPC oncogenesis are not well worked out thus far, and evidence suggests that there may be significant differences between HPV activity in OPC versus cervical cancers. Several changes should be made to the second paragraph in the introduction.
Figure 1 clearly describes the relevance of p16 detection as a surrogate for HPV E7, and would be fine if labelled as such. However, in cervical cancers, E6 and E7 are known to have multiple roles apart from p53/Rb dysregulation. Indeed, their exact roles in OPC tumours is not yet fully worked out.
It is not the expression but the OVEREXPRESSION of E6 and E7 that drives cervical tumour progression. We assume that it is the same for OPC but as the authors point out, HPV-positive OPCs do better than HPV-ve. So other factors may be important.
HPV genome integration is very frequent in cervical cancers but recent evidence has shown that this is not the case in OPCs (see Nulton et al Oncotarget 2017).
A reference for p16 as a surrogate is required in paragraph 3. HPV DNA testing is no longer laborious (line 67).
Line 96 "transcription results in..." is confusing. Perhaps they mean "transcription results in high levels of viral mRNAs"? Line 200.
ddPCR is not more sensitive but is more accurate than standard PCR and a more recent reference to the technique would be helpful.
Reviewer 2 Report
Authors discuss the importance of HNSCC patients stratification according to their HPV status. They discuss advantages and disadvantages of individual analytical methods, including p16 IHC, DNA/RNA ISH, PCR in tissue samples, FNAB, and in blood cells. The review is well written with clear language and therefore easy to read and understand. Without doubt the HPV screening is of great importance as evidenced by fact that it has already been included in TNM v8 revision for oropharynx.
Nevertheless, a significant limitation if this study is in a lack of novelty. There were multiple articles reviewing this topic, eg in TCGA 2015 paper doi:10.1038/nature14129 an integration of multiple HPV-based assays was done with high level of comprehensiveness, so it was widely discussed. So far, this review does not provide new findings (of cited literature just six refs were 2018 and 2019). For instance, a focus to anatomical locations with less-established link to HPV than oropharyngeal tumor would be beneficial.
Article could benefit when unification of systematicity would be performed: in some parts, like for tissue biopsy p16 analysis they mention sensitivity determined in studies as well as pros and cons of this diagnostic test. Such approach should be strictly applied to all other diagnostic tests.
Furthermore, summarizing table should be extended to also include limitations, advantages and apart from sensitivity/specificity more preferably – area under curve as sensitivity/specificity is cutoff-based.
Abstract is not informative for the reader.
Part „4.1. Circulating tumor cells and circulating cell-free tumor DNA“ does not provide any information relevant to HPV analyese.
